# Optical Devices in Tracheal Intubation—State of the Art in 2020

**DOI:** 10.3390/diagnostics11030575

**Published:** 2021-03-22

**Authors:** Jan Matek, Frantisek Kolek, Olga Klementova, Pavel Michalek, Tomas Vymazal

**Affiliations:** 11st Department of Surgery—Department of Abdominal, Thoracic Surgery and Traumatology, First Faculty of Medicine, Charles University in Prague and General University Hospital in Prague, 12800 Prague, Czech Republic; jan.matek@vfn.cz; 2Medical Faculty, Masaryk University, 62500 Brno, Czech Republic; 3Department of Anesthesiology and Intensive Medicine, University Hospital Motol, V Úvalu 84, 15000 Praha, Czech Republic; frantisek.kolek@fnmotol.cz; 4Department of Anesthesiology and Intensive Medicine, University Hospital Olomouc, I.P. Pavlova 185, Nová Ulice, 77900 Olomouc, Czech Republic; olga.klementova@fnol.cz; 5Department of Anesthesiology and Intensive Medicine, First Faculty of Medicine, Charles University in Prague and General University Hospital, U Nemocnice 499/2, 12808 Praha, Czech Republic; pavel.michalek@vfn.cz; 6Department of Anaesthesia, Antrim Area Hospital, Antrim BT41 2RL, UK

**Keywords:** supraglottic airway, optical intubation, optical stylet, hybrid device, flexible fiberoptic intubation, fiberoptic tracheal tubes

## Abstract

The review article is focused on developments in optical devices, other than laryngoscopes, in airway management and tracheal intubation. It brings information on advantages and limitations in their use, compares different devices, and summarizes benefits in various clinical settings. Supraglottic airway devices may be used as a conduit for fiberscope-guided tracheal intubation mainly as a rescue plan in the scenario of difficult or failed laryngoscopy. Some of these devices offer the possibility of direct endotracheal tube placement. Hybrid devices combine the features of two different intubating tools. Rigid and semi-rigid optical stylets represent another option in airway management. They offer benefits in restricted mouth opening and may be used also for retromolar intubation. Awake flexible fiberoptic intubation has been a gold standard in predicted difficult laryngoscopy for decades. Modern flexible bronchoscopes used in anesthesia and intensive care are disposable devices and contain optical lenses instead of fibers. Endotracheal tubes with an incorporated optics are used mainly in thoracic anesthesia for lung separation. They are available in double-lumen and single-lumen versions. They offer a benefit of direct view to the carina and do not require flexible fiberscope for their correct placement.

## 1. Introduction

The historical information about the insertion of various kinds of tubes into the airway to facilitate ventilation can be found from 3600 B.C. in ancient Egypt [1]. Hippocrates identified the cartilage defects as the most serious complication following tracheotomy in about 400 B.C. [2]. The first direct laryngoscopy was performed by a baritone singer, vocal coach, and honorary Doctor of Medicine, Manuel Garcia in 1854 [3]. The pioneer scientific information about tracheal intubation was published at the beginning of the last century. The very first article comes from 1900 and depicts medical decision making in laryngeal stenosis due to diphtheria: “tracheal intubation is considered to be less dangerous than the tracheotomy” [4]. Since 1900 more than 44,000 articles having a “tracheal intubation” as a keyword have been published in the PubMed database. Today, technological advances enable more frequent use of new devices, which may be very useful and safe in various circumstances in the clinical practice of airway management.

## 2. The Purpose of the Review

This review article is focused on the clinical use and innovations of the devices containing optical fibers or lens and portable monitors during the process of tracheal intubation. Most review articles in this field focus on laryngoscopy or video-laryngoscopy. Very few reviews have been published on different optical airway devices and most of them are outdated [5,6,7]. The last decade is characterized by the development of novel medical technologies including high-quality imaging portable devices for the management of the difficult airway. This narrative review strives to contribute by upgrading the evidence on the available optical devices, introducing new advancements related to the design and technologies, and explaining the possibilities when direct laryngoscopy or video-laryngoscopy is not feasible. In the current COVID-19 pandemics era some of the devices described in this review may also theoretically improve the safety of the operator during the process of tracheal intubation by increasing the distance between the patient’s oral cavity and the face of the anesthesiologist or decrease generation of the aerosol.

## 3. Methodology

Following databases were searched: PubMed, Scopus, and Web of Science for an initial search of the articles, and Google Scholar for the additional search of grey literature. The years considered were 1967 (when the first fiberoptic intubation was described)—October 2020. The search did not have any language limitations; however, information for non-English manuscripts were retrieved from the abstracts only. Regarding publication status, all published or accepted manuscripts published online ahead of print were included. Systematic reviews, narrative reviews, randomized controlled trials, cohort studies, case reports, correspondence (letters to the editor) containing description of new cases were further processed. The purpose was to find the highest available evidence for each airway device described in the review (Table 1). Manikin, other simulation, cadaver, or animal studies were not included, as well as the correspondence not describing the original cases. The terms entered included “tracheal intubation”, “supraglottic airway device”, “optical stylet”, “hybrid device”, “fibreoptic/fibreoptic”, and “VivaSight”. Detailed search strategy using the Boolean operators is described in the Appendix A. Two researchers, P.M. and J.M. performed the initial search, T.V. reviewed their results independently. One-hundred and fifteen references were used for the purpose of this review.

## 4. Supraglottic Airway Devices and Optical Intubation

A functional supraglottic laryngeal airway device was invented by Archibald Brain in the early 1980s and in 1987 the first commercial laryngeal mask airway (LMA) was manufactured. The LMA was created to be used as an alternative to the endotracheal tube or the face mask with either spontaneous or positive pressure ventilation [8]. Since it is still widely used today and LMAs have been widespread into daily practice in the operation rooms (OR) and intensive care units (ICU) for both adults and children. A wide variety of original and new generations of specialized LMAs are available nowadays. The revised practice guidelines of the American Society of Anesthesiologists (ASA) for management of difficult airways published in 2013, included use of the LMA or another supraglottic airway device as a rescue device for ventilation and as a conduit for insertion of an endotracheal tube [9]. It can be done either blindly or guided by a fiberoptic bronchoscope; the technique was firstly described in the late 1960s. Fiberoptic-guided tracheal intubation through a supraglottic airway device as a conduit is a technique described as a “Plan B” of the difficult airway guidelines for the management of unanticipated difficult tracheal intubation [10]. This technique can also be used to manage an anticipated difficult airway. Second-generation supraglottic devices incorporate features to minimize the risk of gastric aspiration and are recommended for use in routine clinical practice, as well as to rescue failed tracheal intubation [11]. Fiberscope-assisted intubation via placed supraglottic airway devices has been described as a safe and easy procedure to manage difficult airways. However, visualization of the glottis aperture is essential for any fiberscope-assisted intubation. Various supraglottic airway devices are commercially available and offer different conditions for fiberscope-assisted intubation [5]. The ability to perform tracheal intubation via a supraglottic device can influence the type of device used when managing a difficult airway. The following points should be taken into consideration when selecting the appropriate device: familiarity of the operator with the supraglottic airway device used, features of the device allowing direct fiberscope-guided endotracheal tube placement, and the efficacy of protection against gastric contents aspiration [5,6]. In the current pandemic era, the disposability of the device is also an important factor. There is a number of techniques using introducers or catheters to facilitate the insertion of an appropriate endotracheal tube, particularly guided by a flexible bronchoscope. Usage of introducers or catheters through a supraglottic airway may be as well a useful alternative intubation technique in difficult airway management [5].

### Clinical Evidence

The first generation supraglottic perilaryngeal sealing airway devices have been trialed for both blind and fiberscope-guided tracheal intubation. However, only the Intubating laryngeal mask airway (ILMA, Fastrach) (Laryngeal Mask Company Ltd., Mahé Seychelles, Seychelles), Aura-i laryngeal mask (Ambu A/S, Ballerup, Denmark) [12], and Air-Q Intubating Laryngeal Airway (Mercury Medical, Clearwater, FL, USA) are feasible conduits for fiberoptic tracheal intubation. The ILMA showed a more than 90% success rate for blind insertion of the endotracheal tube, and in all failed blind intubations the subsequent fibreoptic technique was successful [13,14]. Joffe et al. reported a 100% success rate of Air-Q ILA insertion in adults but fiberoptic tracheal intubation succeeded only in 92% of patients [15]. The Air-Q was compared with the Aura-i laryngeal mask, and also with the i-gel in preschool children scheduled for elective surgeries [16,17]. All three supraglottic airway devices were found feasible conduits for fiberscope-guided tracheal intubation, with a high success rate, even if performed by relatively inexperienced operators. The i-gel^®^ (Intersurgical, Berkshire, UK) is a second-generation supraglottic airway device used in both anesthesia and resuscitation [18]. Fiberscope-guided tracheal intubation via the i-gel has been shown to have a high first-attempt success rate and provide a favorable glottic view [19] (Figure 1). Furthermore, the lumen of the device allows direct placement of a reasonably sized endotracheal tube. Fiberscope-guided tracheal intubation through the i-gel achieved a 96% success rate on the first attempt in patients with predicted difficult intubation which was statistically similar to fiberoptic intubation through the single-use Intubating Laryngeal Mask Airway [19]. Some of the supraglottic airway devices have even been specifically designed to facilitate fiberoptic intubation, such as Aura-i [12] or LMA Protector [20]. Fiberoptic intubation through the Aura-i laryngeal mask was compared with the same method through the Ambu AuraGain laryngeal mask in preschool children [21]. The AuraGain device showed a significantly higher success rate of tracheal intubation—100% vs. 79% and exhibited also higher oesophageal seal pressure. Both i-gel supraglottic airway and AuraGain laryngeal mask were found as reliable conduits for fiberoptic intubation in morbidly obese patients, and they provided a good quality view to the vocal cords with effective oxygenation during tracheal intubation [22]. The AuraGain provided better intubation conditions mainly in patients with difficult laryngoscopy than the ILMA laryngeal mask for fiberscope-guided endotracheal tube placement [23]. The Laryngeal Mask Airway Unique (LMA^®^ Unique™, Teleflex Medical, Ireland) and the Laryngeal Mask Airway Aura-once (Ambu^®^ AuraOnce™, Copenhagen, DK) provided in one study better coverage of the glottis as assessed by a fiberscope than Laryngeal Tube, LMA Supreme or the i-gel [24]. However, none of the aforementioned laryngeal masks offer ideal conditions for direct placement of the endotracheal tube. The only available base-of-tongue sealer as a conduit for fibreoptic intubation is disposable Intubating Laryngeal Tube Suction (iLTS-D) (VBM Medical, Sulz, Germany). The first results of its use in clinical practice are not completely encouraging [25].

## 5. Hybrid Devices

These devices possess the features of two or more traditional airway management equipment. The examples include CTrach laryngeal mask airway (The Laryngeal Mask Co., Singapore) and TotalTrack Video Laryngeal Mask (MedComflow S.A., Barcelona, Spain). The LMA CTrach was invented in 2005 and first clinical evaluations were published in 2006 [26,27]. It was a supraglottic airway device with a built-in camera inside the bowl, thus allowing direct visualization of the glottis after insertion with subsequent insertion of the specially designed soft endotracheal tube under continuous direct vision. The device was a very feasible teaching tool in the manikin setting; however, it did not gain broader clinical use due to frequent obstruction of the view due to downfolded epiglottis or contamination of the lens [28]. The CTrach also could fail to secure the airways in patients with hypertrophic lingual tonsil [29] and its use has been also associated with occasional episodes of gastric content aspiration caused probably by the device manipulation around the esophageal entrance [30].

TotalTrack VLM combines the features of a video-laryngoscope and intubating laryngeal mask airway with advanced protection against aspiration of gastric contents [31]. (Figure 2). The device consists of a laryngeal mask with a broad gastric content drainage channel, an internal channel for suctioning of secretions from the perilaryngeal space, an optical fiber attached to the portable monitor, and a side-channel, with a similar shape as video-laryngoscope, designated for insertion of the endotracheal tube and ventilation. The initial clinical evaluation was published in 2015.

### Clinical Evidence

CTrach laryngeal mask was initially studied in 100 patients undergoing elective procedures under general anesthesia [26]. The device provided effective ventilation in all patients and tracheal intubation succeeded in 96% of them. Similar results were published also in another cohort study [27]. A large RCT compared blind intubation through the ILMA vs. insertion of an endotracheal tube through the CTrach on 271 patients [32]. The direct vision of the glottis through the CTrach enabled a higher first-attempt success rate but the time of intubation was longer and, in 8% of patients, the vocal cords could not be visualized using the CTrach. In patients with cervical spine immobilization, tracheal intubation through the CTrach LMA was as successful as the fibreoptic intubation through the ILMA [33], but less successful than using the video-laryngoscope Airtraq [34]. The CTtrach was also found inferior when compared with another video-laryngoscope Glidescope [35]. The ILMA provided significantly better intubation conditions than the CTrach in patients with morbid obesity [36]. Tracheal intubation through the CTrach was significantly shorter in lean than in obese patients in another trial [37]. The CTrach was also compared with direct Macintosh laryngoscopy and provided longer intubation time, required more intubation attempts but allowed better oxygenation and visualization of the glottis [38]. Overall success rate of tracheal intubation with the CTrach in published trials was reported as 98% in patients with normal airways and as 93.7% in those with difficult airways [39].

Only a few studies are available for the TotalTrack VLM device. The first evaluation of the device was performed on 100 patients scheduled for elective procedures under general anesthesia, and the device was able to provide effective ventilation, seal pressures more than 35 cmH_2_0, and an unobscured view of the glottis in all cases [31]. Similar cohort study found 98.5% insertion and 95% intubation success rate [40]. Subsequent studies were performed on obese patients [41,42]. Both studies confirmed that the device is an effective tool for oxygenation, ventilation, and as a conduit for tracheal intubation under direct vision. In a series of eleven patients, the TotalTrack served as a rescue device for oxygenation and tracheal intubation in failed laryngoscopic, videolaryngoscopic, or fiberoptic intubation [43]. Recent large prospective assessment of the device on 300 patients confirmed the previous studies, showing a 100% insertion success rate, 85% success of tracheal intubation at the first attempt, and all patients were intubated through the device after re-positioning [44].

## 6. Optical Stylets

These devices used to facilitate tracheal intubation have been used in clinical practice since the 1980s. The optical stylets consist of thin metal or plastic tube with optical fibers inside the device [7]. Novel optical stylets use modern technologies such as “Complementary Metal Oxide Semiconductor“ (CMOS) video chips at the distal opening of the stylet [45]. The optical stylets may be straight or curved (with the bent tip), rigid, or semirigid. Semirigid devices may have a flexible tip which is manipulated according to the airway anatomy using the handle inside the device or they can be pre-formed prior to tracheal intubation. The essential advantage of an intubation stylet represents its small compact dimension which enables successful tracheal intubation even in an insufficiently opened mouth. The term “optical stylet“ was used for the first time by American endoscopists G. Katz and R. Berci in 1979 [46]. They used a straight rigid metal endoscope with a deployed endotracheal tube in more than one hundred patients for performing and teaching intubation of the trachea. However, the straight design of the stylet was not fully feasible for the cases of difficult laryngoscopy and the device had to be mostly introduced with the aid of Macintosh laryngoscope for compressing the base of the tongue and epiglottis [47]. In this manuscript, the authors also report the use of a very thin diameter stylet for the intubation of neonates and infants. Another step in improving the technique of tracheal intubation with the rigid stylets was made in 1983 when Pierre Bonfils described his technique of tracheal intubation in extremely difficult airway conditions caused by Pierre-Robin sequence [48]. His rigid endoscope (Karl Storz KG, Tuttlingen, Germany) was similar to the previous device but instead of a straight design, it employed a fixed curved distal tip at the angle of 40 degrees. This innovative design allowed not only better access to the anteriorly located larynx but also the application of a novel, retromolar technique of tracheal intubation. Shikani optical stylet (Clarus Medical, Minneapolis, MN, USA) has been available since 1996 and the first results were published in 1999 [49]. The stylet is semimalleable and its tip may be bent according to the upper airway anatomy. It contains glass optical fibers transmitting the image to the eyepiece which can be connected to the monitor. Levitan FPS stylet (Clarus Medical, Minneapolis, MN, USA) (Figure 3) was developed in 2006 as a relatively cheap optical device to assist tracheal intubation in emergency situations [50]. Its design is similar to the Shikani optical stylet but its length is 10 cm shorter. SensaScope (Acutronic Medical Systems AG, Hirzel, Switzerland) has been available since 2006 [51] and upgraded in 2011 [52]. The device is a hybrid endoscope and contains a rigid, S-shaped part and flexible intubating tip.

The novel era of intubating stylets started with advances in technology after 2010. The original stylets were often not friendly to use, were relatively expansive, had a longer learning curve than videolaryngoscopes, and were not used extensively in routine or difficult airway management. New optical stylets do not contain optical fibers but they are supplied with the CMOS video-chips, are malleable, and they usually have a small monitor attached to the handle or allow an attachment of the mobile smartphone device [53]. Trachway intubating stylet (Clarus 30,000-V; Biotronic Instruments Enterprise Ltd., Tai Chung, Taiwan, China) was patented and approved for human use in 2009 [54]. It contains an adjustable semirigid stylet with the light source and camera at the distal tip and an attachable pocket color monitor. Stylet Viu (StyletViu Inc., Costa Mesa, CA, USA) is another affordable light viewing device for airway management. The stylet is available in neonatal, pediatric, and adult versions and also contains a dettachable rotatable pocket monitor. The AinCa video stylet (Anesthesia Associates Inc., San Marcos, CA, USA) (Figure 4) is a relatively cheap reusable, fully malleable optical device with a powerful camera with LED illumination and rechargeable LCD monitor [53]. C-MAC Video Stylet (Karl Storz KG, Tuttlingen, Germany) is a rigid optical stylet with an ajustable flexible tip [55]. UE scope (UE Medical Devices Inc., Newton, MA, USA) contains a pocket monitor which can be used for optical stylet or videolaryngoscope blade [56]. The Intular Scope™ (Medical Park, Seoul, South Korea) is a malleable stylet with a camera on the tip of the device, allowing the attachment of an ordinary smartphone as a monitor [57].

### Clinical Evidence

Bonfils stylet was originally designed for retromolar intubation in children with genetic syndromes [48]. The performance of Bonfils stylet was assessed in 400 consecutive patients [58]. They were intubated using either the midline or retromolar approach. The success rate of tracheal intubation was 99% and difficulties were observed mainly in limited mouth opening, high Body Mass Index, and Cormack-Lehane grades 3 and 4. Bonfils stylet was compared with the intubating laryngeal mask in 80 patients with predictors of difficult laryngoscopy [59]. Time to effective ventilation was shorter in the LMA group while intubation took shorter time and patients experienced less sore throat and hoarseness postoperatively in the Bonfils group. The learning curve of tracheal intubation with the Bonfils stylet is relatively shallow, after 20 intubations most operators are able to intubate tracheas without any difficulties [60]. Intubation with the Bonfils stylet should be beneficial in patients with suspected cervical spine injury because it is associated with significantly lesser cervical spine movements than with direct laryngoscopy [61,62]. Several studies have assessed Bonfils stylet as a device for awake tracheal intubation in patients with predicted difficult airway [63,64,65]. A relatively large number of studies was performed on children. The stylet was compared with flexible fiberoptic intubation in anesthetized children with expected difficult intubation [66] and showed shorter intubation times and better quality of the image. Bonfils fiberscope with the assistance of the laryngoscopic blade was also associated with better intubation conditions than Macintosh laryngoscope or GlideScope Cobalt in children without predicted difficult laryngoscopy [67].

Shikani optical stylet was firstly evaluated in a cohort of 120 adults and children scheduled for ENT procedures [49]. All patients were finally intubated using the device, with more attempts required for those with Cormack and Lehane III and IV grades. The device also proved useful in seven children with difficult or failed conventional laryngoscopy [68]. Cervical spine motions associated with tracheal intubation using Shikani stylet were significantly lower than with Macintosh direct laryngoscopy [69]. Shikani stylet was compared with Macintosh laryngoscopy in 270 patients undergoing cervical spine surgery for spondylosis [70]. Both devices performed without any significant difference. Awake tracheal intubation with Shikani stylet was found as equally successful and faster than awake intubation with a flexible fibrescope in patients with unstable cervical spine [71]. Shikani stylet also provided non-inferior intubating conditions when compared with the GlideScope in simulated difficult airways [72]. The performance of the Shikani stylet was also evaluated in inexperienced operators [73]. In a cohort of 301 intensive care patients, the novices were able to intubate 95.2% of them within two attempts. A new generation of the Shikani stylet with a built-in monitor was compared with Macintosh laryngoscopy in neurosurgical patients [74]. Those intubated with Shikani stylet experienced shorter intubating time and better hemodynamic stability. This device has also been recently compared with awake nasal fiberoptic intubation in patients with predicted difficult intubation due to head and neck cancer [75]. Both devices scored a similarly high success rates but intubation with Shikani stylet was faster.

Only a few studies have been published on the use of Levitan FPS optical stylet. The first cohort study evaluated intubating conditions in 301 consecutive tracheal intubations [76]. In patients, scheduled for elective or emergency surgical procedures, Levitan stylet showed a 99.7% success rate and a 1% incidence of the intraoral trauma. Levitan stylet was compared with the Bonfils rigid fibrescope in a cohort of 324 elective surgical patients [77]. The total failure rate was 5.6% in both groups with a significantly higher failure rate in inexperienced users which demonstrated the importance of familiarity and skill development prior to their introduction to a difficult airway cart. Levitan stylet was also tested in the obese population; it showed a high success rate and similar performance as a videolaryngoscope [78].

An initial clinical study with the SensaScope was performed on 32 patients indicated for routine elective surgical procedures and its performance was compared with direct Macintosh laryngoscopy [51]. SensaScope provided better laryngeal views, and all intubations were completed within 60 s. SensaScope was also used for awake tracheal intubation in 13 patients with a predicted difficult airway [79]. The only difficulties were caused by obstruction of the optics with saliva, mucous plugs, or blood. Kleine-Brueggeney et al. compared SensaScope and Bonfils stylet in 200 patients with a simulated difficult airway and found similar intubation success rates but significantly shorter intubation time and better fiberoptic views in the SensaScope group [80]. They concluded that the flexible tip of Sensascope may offer an additional advantage in assessing the difficult airway. Another study, enrolling only the patients without predictors of difficult airway management, found SensaScope inferior in all parameters of tracheal intubation in comparison with McGrath5 series videolaryngoscope [81].

Trachway video stylet has been trialed in awake orotracheal intubation [82], for optical placement of the left double-lumen tube for thoracic procedures [83] or in simulated cervical spine injuries [84]. Trachway device was compared with the Macintosh laryngoscopy in 150 patients undergoing “rapid sequence induction“ to general anesthesia [85]. Almost all patients were intubated successfully within 60 sec with either device. Lee et al. evaluated Trachway stylet for awake nasotracheal intubation in patients with limited mouth opening [86]. The authors concluded that this technique was easier and faster than awake flexible fibreoptic nasal intubation.

Several authors investigated the possibility that optical stylet intubation may result in less stimulation than direct laryngoscopy and may offer some protective effects from sympathetic hyperactivity. They uniformly concluded that the stylet facilitated tracheal intubation is not better than traditional methods at producing sympathetic stimulation but the time to intubation is shorter and the failure rate is lower in stylet groups [87].

## 7. Flexible Fiberoptic Intubation

The position of awake flexible fiberoptic intubation in the algorithms of difficult airway management is indisputable [88]. First described with the use of a flexible choledoscope in 1967 by Murphy [89], the instruments have significantly improved over the decades. Modern flexible laryngoscopes and bronchoscopes are fully portable, have a battery source of the light. They contain a built-in monitor or allow attachment to a light transportable monitor. Most modern flexible bronchoscopes used in the intensive care are disposable which is extremely important for the reduction of infection transmission in highly infective diseases such as COVID-19. Recently published guidelines support the use of single-use bronchoscopes in both suspected and confirmed COVID-19 patients [90,91]. This applies mainly to bronchoscopic procedures in intubated or tracheostomized patients such as bronchoalveolar lavage, sampling, targeted suctioning, examination for bleeding from the tracheobronchial tree. On the contrary, fiberoptic intubation in COVID-19 patients is not recommended, unless clearly indicated, and videolaryngoscopy is preferred instead mainly due to lesser generation of aerosol and faster duration [92,93]. If fiberoptic intubation is necessary, this should be performed either through the supraglottic airway device [94] or wearing fully protective equipment and a special negative pressure hood [95]. These disposable bronchoscopes do not use optical fibers anymore but instead contain a light-emitting diode (LED) for the illumination and a charge-coupled video chip to transfer the picture on the monitor. The aScope™ 4 (Ambu, Ballerup, Denmark) is a novel disposable flexible bronchoscope, available in three different sizes, and it can be used for difficult intubation, detailed bronchoscopy in the ICU, or even for advanced bronchological evaluations and interventions (Figure 5). This device was preferred over conventional reusable bronchoscopes by anesthesiologists, intensive care physicians, and bronchologists [96]. C-MAC Five S (Karl Storz, Tuttlingen, Germany) is another high quality disposable flexible endoscope used for difficult airway management and a bronchoscopy in the intensive care setting.

### Clinical Evidence

Single-use flexible bronchoscopes and rhinolaryngoscopes have been used routinely over the world mainly during the last year. The older version of disposable fiberscope, aScope 2, was reported as inferior to the conventional reusable flexible bronchoscope for tracheal intubation in patients with a simulated cervical spine injury, mainly because of lower quality of the view and longer intubation time [97]. Similar results were confirmed also in a cohort of patients with predicted difficult airway scheduled for awake tracheal intubation [98]. Another comparison of aScope with Karl Storz conventional bronchoscope did not find any significant differences between these two devices [99]. Cost analysis of disposable versus conventional bronchoscopes in the operating rooms and ICUs showed a higher cost for disposable scopes but their financial benefit in low volume hospitals [100,101]. Bronchoalveolar lavage with the single-use bronchoscopes achieved higher aspirated volumes than with the conventional devices [102]. A novel version of the disposable bronchoscope, aScope 4, was compared with a conventional bronchoscope in 176 patients requiring bronchoscopic intervention [96]. The aScope 4 was preferred in awake intubation, diagnostic, and therapeutic bronchoscopy due to better mobility, maneuverability, and quality of visualization.

## 8. Fiberoptic Tracheal Tubes

Fiberoptic tracheal tubes were created to take advantage of both standard tracheal tubes for airway management and video-laryngoscope for visualization of anatomical structures of the larynx, vocal cords, and proximal parts of the trachea. Since 2012, 34 articles having a “VivaSight” as a keyword have been published in the PubMed database.

VivaSight© (Ambu©, Ballerup, Denmark) is a sterile single-use plastic tube firstly clinically used in 2012. It embodies 2 light-emitting diodes and a miniature video camera. It operates at a high resolution (HD) of 76.800 pixels. VivaSight© provides an 85° diagonal field of vision in 12- to 60-mm depth via the progressive scan mode sensor. The cylinder-shaped camera is 3 mm in diameter and 16 mm in length. The distal aperture of the tube is elliptical shaped, and thus, it becomes possible to accommodate the camera [103]. Two Murphy eyes are positioned bilaterally to the camera. In this way, the laryngeal entrance can be visualized which facilitates intubation of the patients. Power transfers to the video camera and light source via a cable embedded in the wall of the tube. The image is transmitted to a 7″ medical-grade LCD HD monitor (Figure 6). In the presence of intensive secretion, mucus, or blood obscuring the image, the camera lens can be cleansed with 1 to 2 mL liquid via the flushing system; but the efficiency is limited under certain circumstances [104]. Two different types of fiberoptic tracheal tubes are now available: double-lumen tube (DLT) and single-lumen tube (SLT). The SLTs can be supplemented with a bronchial blocker during one-lung ventilation without requiring a fiberoptic bronchoscope [103,105]. The key benefits of the fiberoptic tubes mainly are continuous visualization of the airway, real-time monitoring of proper positioning, and effective intubation which altogether increase patient safety [106,107]. These fiberoptic tubes can be used in both anticipated and unanticipated difficult intubations, lung separation techniques, and one-lung ventilation.

### Clinical Evidence

Randomized clinical trials (RCTs) have proved that the VivaSight DLT camera allowed faster insertion and facilitated initial positioning, with effective lung separation. It also confirmed proper tube positioning intraoperatively and facilitated repositioning when necessary [108]. Patients intubated by the VivaSight DL did not require fiberoptic bronchoscopy during intubation or surgery. The VivaSight DL enables significantly more rapid intubation compared with the conventional double-lumen tube [109]. Another RCTs compared the incidence of fiberoptic bronchoscope (FOB) use during verification of initial placement and for reconfirmation of correct placement following repositioning when either a standard double-lumen tube or fiberoptic double-lumen tube was used for lung isolation during thoracic surgery. This study demonstrated a reduction of 86.8% in FOB use [110]. It was a similar reduction to that found in other published studies [104,105]. There are specific clinical conditions as tracheal and bronchial tumors where the fiberoptic double-lumen tube cannot be used. When using a bronchial blocker, a fiberoptic bronchoscope is required to verify the position of the bronchial blocker, because the repeated use of the fiberoptic bronchoscope increases the risk of tumor rupture and/or hemorrhage. In these situations, the VivaSight© SLT guided bronchial blocker for one-lung ventilation can be used with advantage [111,112]. VivaSight SLT has been compared with Macintosh direct laryngoscopy in critically ill patients requiring tracheal intubation [113]. It did not show any benefit in terms of success rate or speed of intubation.

## 9. Limitations

This article has the general limitations of any narrative review—mainly the possibility of selection and reporting bias. Although the search was performed without language limitation, the data extracted from the non-English manuscripts may have been incomplete. We also did not search for unpublished sources, manuscripts in the review or the trials in progress. This could cause reporting bias because rather the studies with positive results have a tendency to be published. The content of related conference papers was analyzed only randomly using the additional search through the Google Scholar database. We did not have the access to the complex search of the grey literature such as the Northern Light database. The methodological quality of the evidence published in this review has not been graded properly and this can also contribute to the bias.

## 10. Directions for the Future Research

Some airway devices described in this article are backed up by high-quality randomized controlled trials but most of the others rely only on cohort studies or case series. Within the European Union, a medical device may be CE marked without any published high-quality evidence and may be approved for clinical use even if its latter performance is disputable. In the future, national or even international cooperation should be created to test the new airway devices regarding their feasibility and safety. The first step in this process was undertaken when the group of experts from Difficult Airway Society developed and published an initiative for trialing novel supraglottic airway devices—ADEPT [114]. Furthermore, most RCTs on these devices are performed on patients with normal airways or simulated difficult airways usually by application of the cervical collar. Multicenter cooperation will be needed in the future to create proper RCTs in patients with real difficult airway anatomy such as in maxillofacial emergencies or patients with genetic syndromes. Large multicenter cohort studies will be also required to establish a realistic incidence of perioperative and postoperative complications.

## 11. Conclusions

Fiberoptic devices have become popular across anesthesiology and intensive medicine providers during past decades. They take advantage of classical airway management and visualization of the airway anatomy. Fiberoptic devices enable fast and successful intubation in both predictive and unpredictable difficult airways. Awake procedures as video-laryngoscopic assessment or flexible nasal endoscopy can be done prior to surgery. They may prevent emergency tracheostomy in some cases. Due to technical limitations, they are sensitive to saline, blood, or mucus pollution of the optics and obstruction of narrow working channels. In spite of the lower intubation failure rate in novice users, adequate skill development prior to their introduction into routine airway management is necessary. In general, advances in tracheal intubation during the last years represent mainly improvement in visualization, easy handling, and safe management of the airway for all [115]. In the current coronavirus era, one of the most important parameters also becomes the disposability of the devices.

## Figures and Tables

**Figure 1 diagnostics-11-00575-f001:**
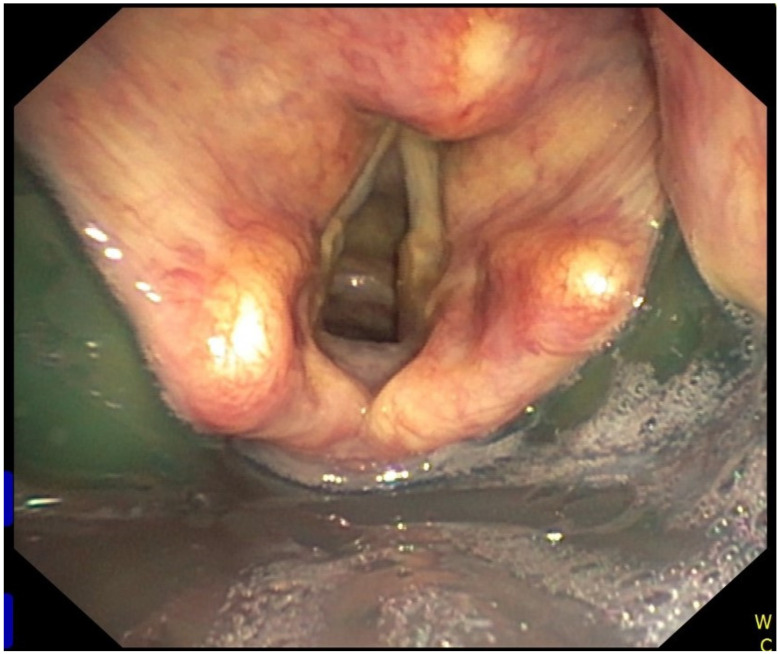
Fiberoptic intubation through the i-gel supraglottic airway.

**Figure 2 diagnostics-11-00575-f002:**
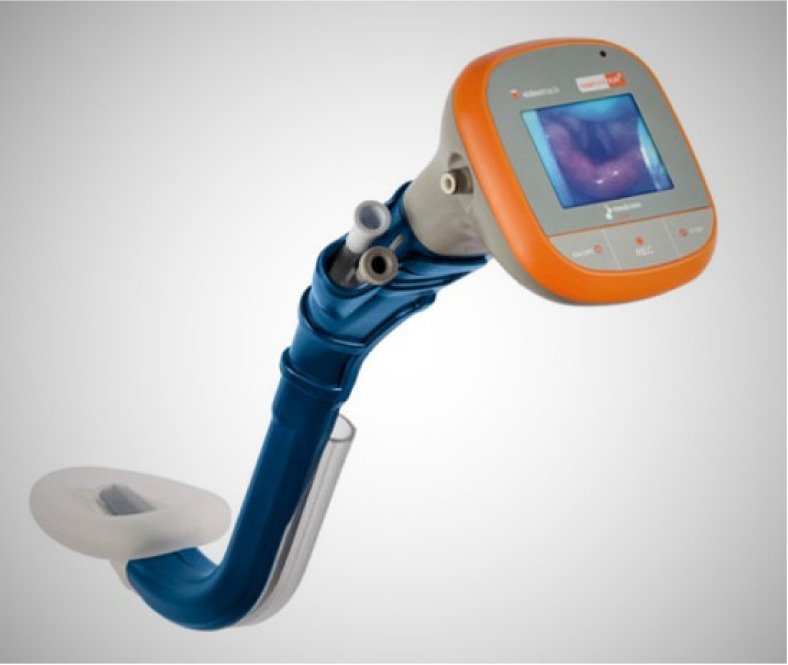
TotalTrack video laryngeal mask.

**Figure 3 diagnostics-11-00575-f003:**
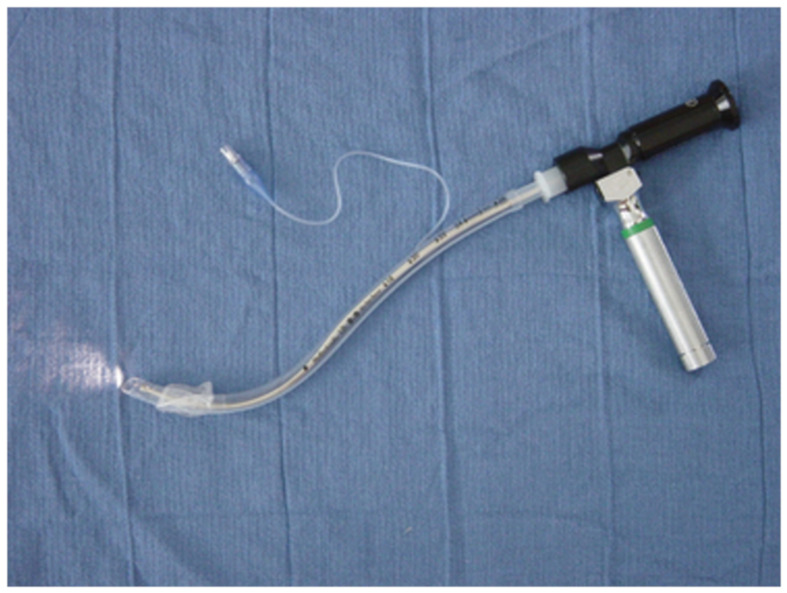
The Levitan FPS optical stylet inserted in a single lumen tracheal tube.

**Figure 4 diagnostics-11-00575-f004:**
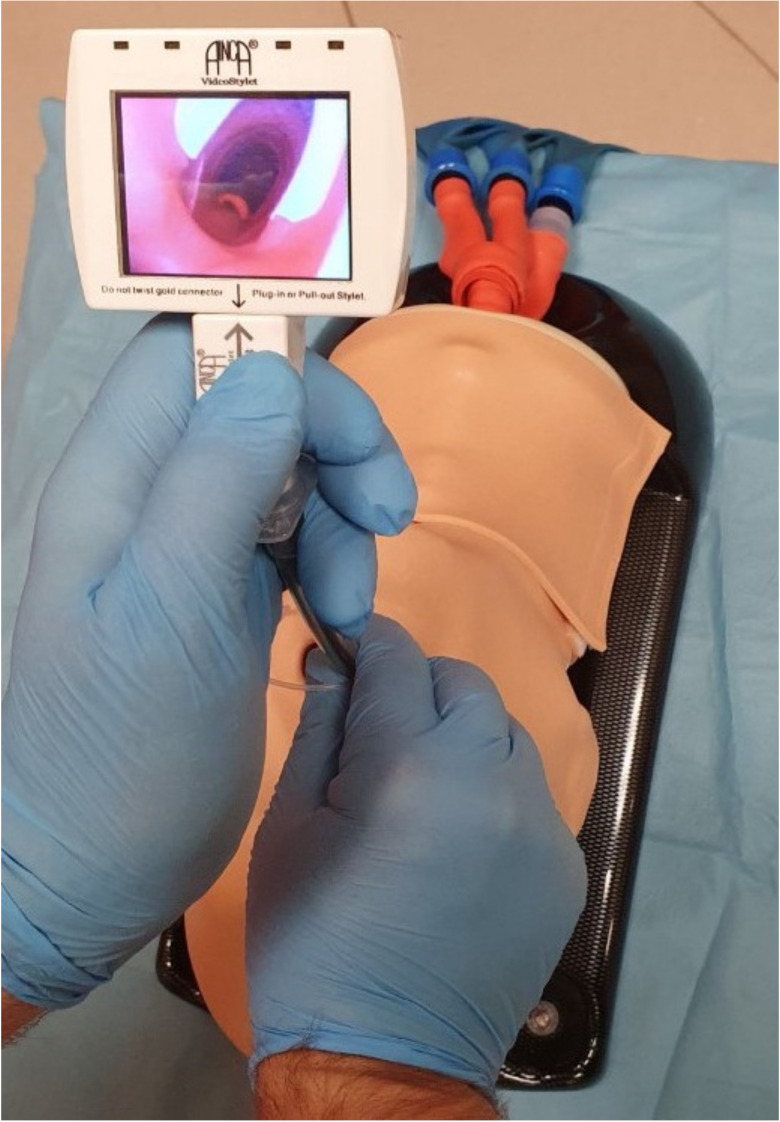
AinCa intubating video stylet.

**Figure 5 diagnostics-11-00575-f005:**
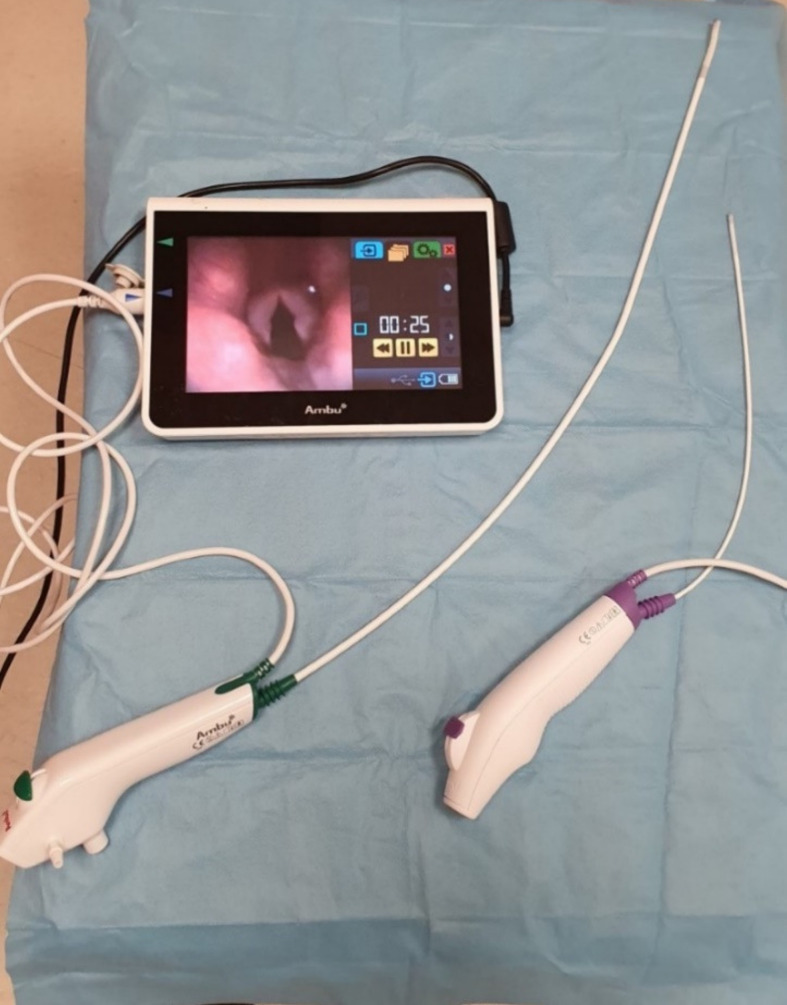
Ambu aScope 4 flexible bronchoscopes.

**Figure 6 diagnostics-11-00575-f006:**
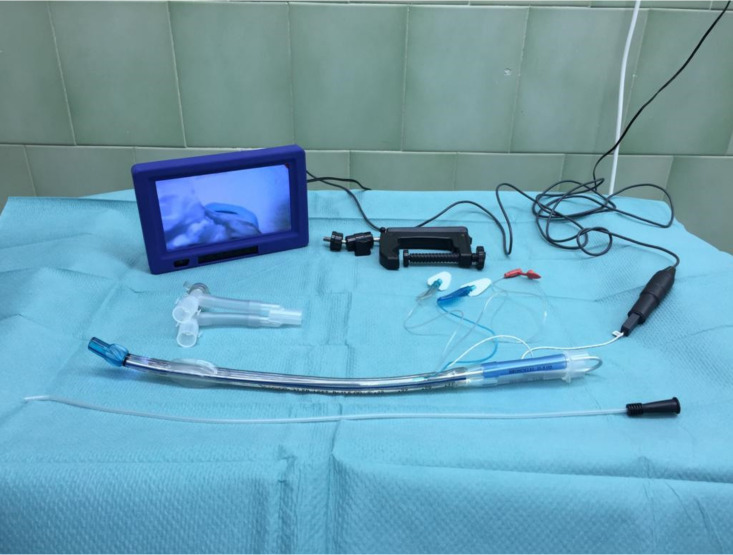
The fiberoptic double-lumen tracheal tube.

**Table 1 diagnostics-11-00575-t001:** Optical devices used for tracheal intubation and their highest available level of evidence.

Device Type	Device	Manufacturer	Level of Evidence
Supraglottic airway	ILMA, Fastrach	Laryngeal Mask Co., Mahé, Seychelles	1b
	Air-Q ILA	Mercury Medical, Clearwater, FL, USA	2b
	i-gel	Intersurgical Ltd., Wokingham, UK	1b
	Aura-i	Ambu A/S, Ballerup, Denmark	2b
	AuraGain	Ambu A/S, Ballerup, Denmark	2b
	LMA Protector	Teleflex Medical Ltd., Limerick, Ireland	2b
Hybrid device	LMA CTrach	The Laryngeal Mask Co., Singapore	1a
	TotalTrack VLM	MedComflow S.A., Barcelona, Spain	2b
Optical stylet	Bonfils stylet	Karl Storz KG, Tuttlingen, Germany	1a
	Shikani stylet	Clarus Medical, Minneapolis, MN, USA	2b
	Levitan FPS stylet	Clarus Medical, Minneapolis, MN, USA	2b
	Sensascope	Acutronic AG, Bubikon, Switzerland	2b
	Trachway,	Biotronic Instruments Ltd., Beijing, China	2b
	Stylet Viu	StyletViu Inc., Costa Mesa, CA, USA	5
	AinCa video stylet	Anesthesia Associates Inc., San Marcos, CA, USA	5
	C-MAC Video Stylet	Karl Storz KG, Tuttlingen, Germany	4
	UE Scope Video Stylet	UE Medical Devices Inc., Newton, MA, USA	5
	Intular Scope	Medical Park, Sinsu-ro, South Korea	2b
Flexible endoscope	aScope	Ambu A/S, Ballerup, Denmark	2b
(single-use)	C-MAC Five S	Karl Storz KG, Tuttlingen, Germany	5
Optical tracheal tube	VivaSight-SL	Ambu A/S, Ballerup, Denmark	2b
	VivaSight-DL	Ambu A/S, Ballerup, Denmark	2b

Legend: 1a—systematic review of (homogenous) randomized controlled trials, 1b—individual randomized controlled trials (with narrow confidence intervals), 2a—systematic review of (homogenous) cohort studies of “exposed” and “unexposed” subjects, 2b—individual cohort studies or low-quality randomized controlled trials, 3a—systematic review of (homogenous) case-control studies, 4—case series, low-quality cohort or case-control studies, 5—expert opinions based on non-systematic reviews of results or mechanistic studies (https://guides.library.stonybrook.edu/evidence_based_medicine/levels_of_evidence/).(accessed on 20 February 2021).

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
