# Peer review of "Optical Devices in Tracheal Intubation—State of the Art in 2020"

_diagnostics, 2021, doi:10.3390/diagnostics11030575_

Round 1
Reviewer 1 Report
I read this paper with interest. The authors focused on optical devices, other than laryngoscopes, and they summarized the advantages, limitations, and clinical applications of various devices very well.
My comments are as follows:
It seems that the methodology section can be expanded more specifically.
There are some errors in the text. I recommend that the authors proofread the manuscript carefully. Here are some examples of where corrections may be required.
Abstract
Line 29. "They are available in a double-lumen a single-lumen version."
Main text
Line 230 "C230 MAC Video Stylet (Karl Storz KG, Tuttlingen, Germany) [49]." : This sentence may be incomplete.
Line 232 "The Intular Scope TM (Medical Park, Seoul, South Korea) is a malleable stylet with a camera on the tip of the device, allowing the attachment of an ordinary smartphone as a monitor 234 [51]." : Please check the reference.
Author Response
Reviewer No. 1:
I read this paper with interest. The authors focused on optical devices, other than laryngoscopes, and they summarized the advantages, limitations, and clinical applications of various devices very well.
My comments are as follows:
It seems that the methodology section can be expanded more specifically.
A: we have expanded methodology section appropriately, the recommendations of the Reviewer No. 3 were accepted. (see below)
There are some errors in the text. I recommend that the authors proofread the manuscript carefully. Here are some examples of where corrections may be required.
Abstract
Line 29. "They are available in a double-lumen a single-lumen version."
A: Corrected: "They are available in double-lumen and single-lumen versions.”
Main text
Line 230 "C230 MAC Video Stylet (Karl Storz KG, Tuttlingen, Germany) [49]." : This sentence may be incomplete.
A: Corrected: C-MAC Video Stylet (Karl Storz KG, Tuttlingen, Germany) is a rigid optical stylet with an adjustable flexible tip [49].
Line 232 "The Intular Scope TM (Medical Park, Seoul, South Korea) is a malleable stylet with a camera on the tip of the device, allowing the attachment of an ordinary smartphone as a monitor 234 [51]." : Please check the reference.
A: correct reference has been added: Lee, J.Y.; Hur, H.J.; Park, H.Y.; Jung, W.S.; Kim, J.; Kwak, H.J. Comparison between video-lighted stylet (Intular ScopeTM) and direct laryngoscope for endotracheal intubation in patients with normal airway. J. Int. Med. Res. 2020, 48, 300060520969532.
Reviewer 2 Report
This article comprehensively review the usefulness of optic device in tracheal intubation. Overall, the article is well-written and provide useful information regarding this issue. I just have one minor suggestion.
1. Please add a table to make a brief summary.
Author Response
Reviewer No. 2:
This article comprehensively review the usefulness of optic device in tracheal intubation. Overall, the article is well-written and provide useful information regarding this issue. I just have one minor suggestion.
- Please add a table to make a brief summary.
A: A table summarizing the devices including the evidence available for their use has been added.
Reviewer 3 Report
I review the article entitled "Optical Devices in Tracheal Intubation – state of the art in 2020" by Matek et al. The review article sheds the light on developments in optical devices in airway management and tracheal intubation. The authors introduce key existing literature and summarizing the current state of the field. I feel in several places, this review could be strengthened. My comments regarding this review are listed below:
Introduction
1
Describe the rationale for the review more clearly. How does this review contribute to literature? Why this review is needed? Please explain why a review of the topic is necessary in the era of COVID-19 pandemic. In other word, please define why did you focused on the clinical use of and innovations of the devices containing optical fibers or lens and portable monitors during the process of tracheal intubation.
2
Following three descriptions require appropriate citations:
Lines 35-36
"The historical information..3600 B.C. in ancient Egypt"
Lines 36-38
Hippocrates identified... 400 B.C.
Lines 38-39
The first direct laryngoscopy was performed by a baritone singer, vocal coach.. .Manuel Garcia in 1854.
Methodology
A well-constructed search strategy is the core of your review. The reviewer thinks several information is missing in the current form.
3
Specify report characteristics (e.g., years considered, language, publication status) used as criteria for eligibility, giving rationale.
4
When this literature search was conducted? Who conducted this literature search? How many articles were extracted from PubMed, Web of Science, Scopus, and Google scholar? How many articles were excluded due to the dupulicate?
5
The flow diagram showing the number of records identified, included and excluded, and the reasons for exclusions should be provided. The reviewer understands this is a narrative review and not a systematic review, but PRISMA 2009 Flow Diagram (http://prisma-statement.org/prismastatement/flowdiagram) is useful to address this comment.
6
Did you combine search terms using Boolean Operators (e.g. OR, AND NOT)? Search formula should describe in detail.
Clinical evidence of each device
7
Level of Evidence should be provided.
https://guides.library.stonybrook.edu/evidence-based-medicine/levels_of_evidence
8
Do you have any implications for future research? Please identify gaps in existing studies for potential future research.
9
Performing bronchoscopy or fiberoptic bronchoscopy-assisted endotracheal intubation in times of the Covid-19 pandemic (e.g. significance, trend, safety) should be described more in detail.
10
Discuss limitations of your review-level (e.g., incomplete retrieval of identified research, reporting bias).
Funding
11
Please describe sources of funding for this review and role of funders for this review.
Author Response
Reviewer No. 3:
I review the article entitled "Optical Devices in Tracheal Intubation – state of the art in 2020" by Matek et al. The review article sheds the light on developments in optical devices in airway management and tracheal intubation. The authors introduce key existing literature and summarizing the current state of the field. I feel in several places, this review could be strengthened. My comments regarding this review are listed below:
Introduction
1
Describe the rationale for the review more clearly. How does this review contribute to literature? Why this review is needed? Please explain why a review of the topic is necessary in the era of COVID-19 pandemic. In other word, please define why did you focused on the clinical use of and innovations of the devices containing optical fibers or lens and portable monitors during the process of tracheal intubation.
A:
Text added to the manuscript:
This review article is focused on the clinical use and innovations of the devices containing optical fibers or lens and portable monitors during the process of tracheal intubation. Most review articles in this field focus on laryngoscopy or videolaryngoscopy. Very few reviews have been published on different optical airway devices and most of them are outdated [5-7]. The last decade is characterized by the development of novel medical technologies including high-quality imaging portable devices for the management of the difficult airway. This narrative review strives to contribute by upgrading the evidence on the available optical devices, introducing new advancements related to the design and technologies, and explaining the possibilities when direct laryngoscopy or video-laryngoscopy is not feasible. In the current Covid-19 pandemics era some of the devices described in this review may also theoretically improve the safety of the operator during the process of tracheal intubation by increasing the distance between the patient´s oral cavity and the face of the anesthesiologist or decrease generation of the aerosol.
2
Following three descriptions require appropriate citations:
Lines 35-36
"The historical information..3600 B.C. in ancient Egypt"
A: The reference added: Pahor, A.L. Ear, nose, and throat in Ancient Egypt. J. Laryngol. Otol. 1992, 106, 773-779.
Lines 36-38
Hippocrates identified... 400 B.C.
A: The reference added: Szmuk, P., Ezri, T., Evron, S., Roth, Y., Katz, J. A brief history of tracheostomy and tracheal intubation, from the Bronze Age to the Space Age. Intensive Care Med. 2008, 34, 222-228.
Lines 38-39
The first direct laryngoscopy was performed by a baritone singer, vocal coach.. .Manuel Garcia in 1854.
A: The reference added: Garcia, M. Observation on the human voice. Laryngoscope 1905, 15, 185-194.
Methodology
A well-constructed search strategy is the core of your review. The reviewer thinks several information is missing in the current form.
3
Specify report characteristics (e.g., years considered, language, publication status) used as criteria for eligibility, giving rationale.
A: Following databases were searched: PubMed, Scopus, Web of Science for an initial search of the articles, and Google Scholar for the additional search of grey literature. The years considered were 1967 (when the first fiberoptic intubation was described) – October 2020. The search did not have any language limitations, however information for non-English manuscripts were retrieved from the abstracts only. Regarding publication status, all published or accepted manuscripts published online ahead of print were included. Systematic reviews, narrative reviews, randomized controlled trials, cohort studies, case reports, correspondence (letters to the editor) containing description of new cases were further processed. The purpose was to find the highest available evidence for each airway device described in the review. Manikin, other simulation, cadaver or animal studies were not included, as well as the correspondence not describing original cases.
4
When this literature search was conducted? Who conducted this literature search? How many articles were extracted from PubMed, Web of Science, Scopus, and Google scholar? How many articles were excluded due to the dupulicate?
A: The years considered for the search were 1967 (when the first fiberoptic intubation was described) – October 2020. For the historical introduction, the search was not limited by any time period. Detailed search strategy using the Boolean operators is described in the Appendix. Two researchers, P.M. and J.M. performed the initial search, T.V. reviewed their results independently.
|
Records identified through database search |
Records after duplicates removed |
Records excluded |
Records included |
Supraglottic airway devices
Hybrid devices
Optical stylets
Single-use fibroscopes
Fiberoptic tracheal tubes |
PubMed 200 Scopus 95 Web of Science 63 Google Scholar Other sources 10 Total 368 PubMed 76 Scopus 120 Web of Science 99 Google Scholar Other sources 0 Total 295 PubMed 114 Scopus 109 Web of Science 73 Google Scholar Other sources 30 Total 326 PubMed 25 Scopus 48 Web of Science 31 Other sources 2 Google Scholar Total 106 PubMed 33 Scopus 23 Web of Science 45 Google Scholar Other sources 1 Total 101 |
Total 248
Total 139
Total 212
Total 59
Total 55 |
Not relevant 122 Other reasons 108 Total 230
Not relevant 47 Other reasons 73 Total 120
Not relevant 25 Other reasons 142 Total 167
Not relevant 15 Other reasons 35 Total 50
Not relevant 12 Other reasons 31 Total 43 |
Total 18
Total 19
Total 45
Total 9
Total 12 |
5
The flow diagram showing the number of records identified, included and excluded, and the reasons for exclusions should be provided. The reviewer understands this is a narrative review and not a systematic review, but PRISMA 2009 Flow Diagram (http://prisma-statement.org/prismastatement/flowdiagram) is useful to address this comment.
A: The PRISMA 2009 Flow Diagram using the numbers from Table (Appendix) is provided in the Appendix.
6
Did you combine search terms using Boolean Operators (e.g. OR, AND NOT)? Search formula should describe in detail.
A: Search formula is described in detail in the Appendix.
Clinical evidence of each device
A: Highest available Level of Evidence is provided for each device in Table 1
Table 1. Highest available Level of Evidence for each airway device. Legend: 1a – systematic review of (homogenous) randomized controlled trials, 1b – individual randomized controlled trials (with narrow confidence intervals), 2a – systematic review of (homogenous) cohort studies of "exposed" and "unexposed" subjects, 2b – individual cohort studies or low-quality randomized controlled trials, 3a – systematic review of (homogenous) case-control studies, 4 – case series, low-quality cohort or case-control studies, 5 – expert opinions based on non-systematic reviews of results or mechanistic studies (https://guides.library.stonybrook.edu/evidence_based_medicine/levels_of_evidence/)
Device type |
Device |
Manufacturer |
Level of evidence |
Supraglottic airway
Hybrid device
Optical stylet
Flexible endoscope (single-use) Optical tracheal tube |
ILMA, Fastrach Air-Q ILA i-gel Aura-i AuraGain LMA Protector LMA CTrach TotalTrack VLM Bonfils stylet Shikani stylet Levitan FPS stylet Sensascope Trachway, Stylet Viu AinCa video stylet C-MAC Video Stylet UE Scope Video Stylet Intular Scope aScope C-MAC Five S VivaSight-SL VivaSight-DL |
Laryngeal Mask Co., Seychelles Mercury Medical, USA Intersurgical Ltd., UK Ambu A/S, Denmark Ambu A/S, Denmark Teleflex Medical Ltd., Ireland The Laryngeal Mask Co., Singapore MedComflow S.A., Spain Karl Storz KG, Germany Clarus Medical, USA Clarus Medical, USA Acutronic AG, Switzerland Biotronic Instruments Ltd., China StyletViu Inc., USA Anesthesia Associates Inc., USA Karl Storz KG, Germany UE Medical Devices Inc., USA Medical Park, South Korea Ambu A/S, Denmark Karl Storz KG, Germany Ambu A/S, Denmark Ambu A/S, Denmark |
1b 2b 1b 2b 2b 2b 1a 2b 1a 2b 2b 2b 2b 5 5 4 5 2b 2b 5 2b 2b |
7
Level of Evidence should be provided.
https://guides.library.stonybrook.edu/evidence-based-medicine/levels_of_evidence
A: Level of evidence according to the recommended Levels of Evidence (according to EBM - Straus, S.E. et al. Evidence-based medicine: how to practice and teach EBM. Elsevier, 2018)
8
Do you have any implications for future research? Please identify gaps in existing studies for potential future research.
A: Some airway devices described in this article are backed up by high-quality randomized controlled trials but most of the others rely only on cohort studies or case series. Within the European Union, a medical device may be CE marked without any published high-quality evidence and may be approved for clinical use even if its latter performance is disputable. In the future, national or even international cooperation should be created to test the new airway devices regarding their feasibility and safety. The first step in this process was undertaken when the group of experts from Difficult Airway Society developed and published an initiative for trialing novel supraglottic airway devices – ADEPT (Pandit JJ, et al. Anaesthesia 2011, 66, 726-737) Furthermore, most RCTs on these devices are performed on patients with normal airways or simulated difficult airways usually by application of the cervical collar. Multicenter cooperation will be needed in the future to create proper RCTs in patients with real difficult airway anatomy such as in maxillofacial emergencies or patients with genetic syndromes. Large multicenter cohort studies will be also required to establish a realistic incidence of perioperative and postoperative complications.
9
A: Performing bronchoscopy or fiberoptic bronchoscopy-assisted endotracheal intubation in times of the Covid-19 pandemic (e.g. significance, trend, safety) should be described more in detail.
This is a controversial issue and beyond the scope of this narrative review. In Covid-19 infected and potentially infected patients other techniques – e.g. videolaryngoscopy – wearing full protective equipment (FFP3 mask, helmet, overall, double gloves) are recommended as the first method of choice and awake fibreoptic intubation should not be performed unless strictly indicated (Orser BA. Recommendations for endotracheal intubation of Covid-19 patients. Anesth. Analg. 2020, 130, 1109-1110.; Cook TM et al. Consensus guidelines for managing the airway in patients with Covid-19. Anaesthesia. 2020, 75, 785-799). If fiberoptic intubation indicated, this should probably be performed through the supraglottic airway device (Sorbello M, et al. Prevention is better than the cure, but the cure cannot be worse than the disease: fibreoptic tracheal intubation in Covid-19 patients. Br. J. Anaesth. 2020, 125, E187-E188). However, there are some patients whom must be intubated with the fiberscope such as acute maxillofacial emergencies – there a special hood with a negative pressure may decrease the risk of aerosol creation (Emery AR et al. A novel approach to fiberoptic intubation in patients with coronavirus disease 2019. J. Oral Maxillofac. Surg. 2020, 78, 2182e1-2182e6).
During bronchoscopic procedures in intubated or tracheostomied patients in the ICU, such as bronchoalveolar lavage (BAL), targeted suctioning and toilet of the tracheobronchial tree, apart from the use of FFP3, the use of single-use bronchoscopes in recommended (Barron SP, Kennedy MP. Single-use (disposable) flexible bronchoscopes: the future of bronchoscopy? Adv. Ther. 2020, 37, 4538-4548.; Pritchett MA, et al. Society for Advanced Bronchoscopy Consensus Statement and Guidelines for bronchoscopy and airway management amid the COVID-19 pandemic. J. Thorac. Dis. 2020, 12, 1781-1798)
10
Discuss limitations of your review-level (e.g., incomplete retrieval of identified research, reporting bias).
A: This article has the general limitations of any narrative review – mainly the possibility of selection and reporting bias. Although the search was performed without language limitation, the data extracted from the non-English manuscripts may have been incomplete. We also did not search for unpublished sources, manuscripts in the review or the trials in progress. This could cause reporting bias because rather the studies with positive results have a tendency to be published. The content of related conference papers was analyzed only randomly using the additional search through the Google Scholar database. We did not have the access to the complex search of the grey literature such as the Northern Light database. The methodological quality of the evidence published in this review has not been graded and this can also contribute to the bias.
Funding
11
Please describe sources of funding for this review and role of funders for this review.
This review was funded by the Czech Ministry of Health, grant number MZCZ-DRO-VFN64165. The APC was funded by the research grant of University Hospital in Motol, Prague.
Round 2
Reviewer 3 Report
The revision is satisfactory. I have no further comment.